# Silicon-Selenium Interplay Imparts Cadmium Resistance in Wheat through an Up-Regulating Antioxidant System

**DOI:** 10.3390/ijms25010387

**Published:** 2023-12-27

**Authors:** Maria Manzoor, Muna Ali Abdalla, Md Arif Hussain, Karl Hermann Mühling

**Affiliations:** Institute of Plant Nutrition and Soil Science, Kiel University, Hermann-Rodewald-Street 2, 24118 Kiel, Germany; mabdalla@plantnutrition.uni-kiel.de (M.A.A.); ahussain@plantnutrition.uni-kiel.de (M.A.H.)

**Keywords:** cadmium, selenium, silicon, wheat, glutathione, sulfur, toxicity

## Abstract

Cadmium (Cd), being a highly toxic heavy metal, significantly impacts plant growth and development by altering nutrient uptake and causing oxidative and structural damage, resulting in reduced yield. To combat Cd toxicity and accumulation in wheat, it was hypothesized that co-application of Selenium (Se) and Silicon (Si) can reduce the adverse effect of Cd and regulate Cd resistance while improving Se fortification in wheat. Therefore, this study evaluated the comparative effect of Se and Si on the growth and antioxidant defense systems of wheat plants grown in a hydroponic setup. Briefly, the plants were acclimatized to the hydroponic solution for 1 week and then exposed to 10 µmol Cd. Afterwards, the treatments, including 0.2 mmol Si and 1.5 µmol Se, were applied as a root and foliar application, respectively. Plants supplemented with both Se and Si showed improved biomass and other physiological growth attributes, and this response was associated with improved activity/contents of antioxidants, including glutathione (GSH) content, glutathione reductase (GR), ascorbate peroxidase (APX), and catalase (CAT), with related lowering of hydrogen peroxide, malondialdehyde content, and structural damages. Moreover, by Se + Si supplementation, a decrease in total S levels in plant tissues was observed, whereas an increase in total protein concentration and GSH indicated a different and novel mechanism of Cd tolerance and S homeostasis in the plant. It was observed that Si was more involved in significantly reducing Cd translocation by stabilizing Cd in the root and reducing its content in the soluble fraction in both the root and shoot. Whereas Se was found to play the main role in reducing the oxidative damage caused by Cd, and the effect was more profound in the shoot. In addition, this study also observed a positive correlation between Si and Se for relative uptake, which had not been reported earlier. Our findings show that the Se and Si doses together benefit growth regulation and nutrient uptake; additionally, their combinations support the Cd resistance mechanism in wheat through upregulation of the antioxidant system and control of Cd translocation and subcellular distribution, ultimately contributing to the nutritional quality of wheat produced. Thus, it is concluded that the co-application of Se and Si has improved the nutritional quality while reducing the Cd risk in wheat and therefore needs to be employed as a potential strategy to ensure food safety in a Cd-contaminated environment.

## 1. Introduction

Cadmium is one of the most toxic heavy metals in surface soil that adversely affects soil microbial health, plants, and, subsequently, animals and humans [1]. The natural concentration of Cd in natural agricultural soils is between 0.01 and 1 mg/kg, with a worldwide average of 0.36 mg/kg [2]. Agricultural soils containing Cd > 1 mg/kg are unsuitable for plant growth [3]. Unfortunately, many agricultural soils worldwide have been found to be contaminated with Cd. A potentially high concentration of Cd has also been reported in Germany, France (16.6 mg/kg), Pakistan, and China [4]. Cd, once it enters the soil, adversely affects soil microbial communities by inhibiting cell division, damaging DNA, and impairing functional groups [1]. In plants, Cd results in the inhibition of normal cell division, impaired ion homeostasis, reduced photosynthetic efficiency, production of reactive oxygen species (ROS), reduced mitotic index, and inducing genotoxicity with a risk of Cd exposure to humans through food chain contamination [3,4,5,6,7]. In human beings, Cd may cause injuries, including prostate cancer, bone deformities, and DNA abnormalities [5]. The European Union (EU) has set a limit of 0.2 mg/kg for Cd in foodstuffs (Commission Regulation EC 1881/2006) [8]. The European Food Safety Authority reported that one-third of the European population has an average daily intake of Cd above the recommended maximum (2.5 μg/kg body weight per week), of which one-third of the Cd intake comes directly from the consumption of grain products, including wheat [9]. Many studies have indicated high Cd content in wheat grains (0.01–0.9 mg/kg), above the 0.2 mg/kg for Cd in wheat [2,10]. Wheat (*Triticum aestivum* L.) is a staple food (60% of the world’s population) and a major source of caloric intake globally. Therefore, reducing Cd in wheat grains is vital to ensure food safety and decrease human health risks. To avoid food chain contamination with Cd, using selenium (Se) and silicon (Si) has received much importance in recent years [11].

Se is an essential micronutrient that has antioxidant, anti-cancer, and anti-viral properties [12] and is therefore highly important for humans and animals for a well-balanced immune response. It is reported that 15% of the world population is deficient in Se [13,14]. In addition to being a beneficial nutrient for humans, Se also serves to combat Cd toxicity in plants by regulating glutathione (GSH) peroxidase and reductase activities [15], repairing damaged cells, and reducing oxidative damages, hence contributing to the Cd tolerance mechanism in plants [11]. GSH, an important non-enzymatic antioxidant from the ascorbic acid-glutathione (ASA-GSH) system, is a low-molecular-weight thiol compound that is crucial for reducing the buildup of reactive oxygen species and maintaining redox status homeostasis. GSH also participates in phytochelatin synthesis (PCs), which binds to Cd in the cytosol and transports it to vacuoles [16,17]. On the other hand, silicon (Si) fertilizers are also known for promoting plant growth, nutrition, and metabolism and protecting them against biotic and abiotic stresses and therefore appear to be a promising tool. Si regulates morphological changes in root cells, particularly modulating the cell wall (cellulose, pectin, and lignin) and endodermis (lignin and suberin deposition) [18], therefore contributing to the plant Cd tolerance mechanism. A recent molecular investigation revealed that Si decreases apoplastic Cd translocation to above-ground tissue in winter wheat by up-regulating the expression of TaTM20 (the Cd efflux gene) [18] and increasing root tips and oxalate exudation [19], which decreases Cd transport and translocation in the plant. Despite the potential of Se and Si in reducing the Cd uptake ability of wheat, there is a scarcity of studies that have investigated them together to achieve Cd resistance in wheat. Previous studies have mainly focused on the comparative effect of Si and Se on wheat growth under Cd stress [11,18,19]. To date, few studies have combined Si and Se to combine tolerance and avoidance mechanisms to achieve Cd resistance in wheat, and in most cases, soil experiments were conducted that did not ensure the exact treatment effects of Si and/or Se. This study seeks to fill this gap by examining the spatiotemporal transport process of Cd at root/shoot interfaces and its subcellular distribution as impacted by Se and Si. In this study, we hypothesized that (a) Si-Se co-application may have a synergistic effect on reducing Cd transport and induced toxicity owing to having shared attributes of regulating plant defense mechanisms and stress tolerance; (b) Se may decrease total S and sustain/increase reduced S (GSH) in roots and/or shoots; and (c) explore the interaction of Se and Si for wheat uptake. The research findings will provide novel insights for promoting wheat growth and production in Cd-contaminated environments and a theoretical background for further research development in optimizing Cd resistance in wheat.

## 2. Results

### 2.1. Effects of Se-Si-Cd on Wheat Physiological Growth

Cd exposure significantly reduced plant biomass and shoot length by 1.1– fold (Table 1). Which was improved significantly upon Se and Si treatments. Comparatively highest significant (*p* < 0.05) increase in biomass and shoot length by 1.4– and 1.2– folds were observed in Cd-exposed plants applied with Se + Si treatment followed by Si treatment (1.3– and 1.2– folds), respectively. No significant effect of Cd was observed on a number of tillers; however, upon co-application of Se + Si in Cd-stressed plants, a significant (*p* < 0.05) increase in No. of tillers by 1.3– folds was observed. Plant photosynthetic parameters also indicated Cd-induced toxicity. Results of photosynthesis activity and transpiration indicated a significant reduction of 1.2– fold in Cd-stressed plants. Which were significantly increased by 1.3– and 1.4– folds, respectively. Chlorophyll content was also significantly decreased upon Cd exposure (1.2– folds), which was significantly increased upon Se + Si treatments by 1.2– fold, respectively, as compared to Cd-stressed plants (Table 1).

### 2.2. Cd Uptake Characteristics in Different Plant Components

#### 2.2.1. Cd Concentration in Root/Shoot

Wheat plants exposed to 10 µM had average Cd concentrations of 65.4 and 840.8 mg kg^−1^ in shoot and root, respectively. Both Si and Se significantly decreased the Cd concentration in the root and shoot of wheat. Compared to Se, plants treated with 0.2 mM Si significantly reduced Cd concentration in shoot and root by 1.6– and 1.8– folds, respectively. However, a higher significant reduction was observed when both Se and Si were applied together, and a synergistic response in Cd reduction by 2.1– and 2.6– folds was recorded in shoot and root, respectively. Results from Cd translocation to different plant components indicated that Cd accumulation in the root (840.8 mg kg^−1^) was higher than in the shoot (65.4 mg kg^−1^). In the shoot, more Cd was accumulated in older leaves (30–65 mg kg^−1^) than in younger leaves (20–25 mg kg^−1^).

#### 2.2.2. Subcellular Distribution of Cd

The results from subcellular Cd distribution indicated that in younger leaves, Si and Se addition remarkably decreased Cd concentrations in CW, SF, and OF fractions, as reflected by a significant decrease in total Cd concentration (Figure 1b). In old leaves, a significant decrease in total Cd was attributed to the significant reduction of Cd in the SF and CW fractions, whereas no significant difference was observed in the OF fraction regardless of Si/Se addition (Figure 1c). In roots, Se and Si supplementation both significantly (*p* < 0.05) reduced the total Cd concentration (Figure 1a–d). With regard to subcellular distribution proportion, it was observed that Se substantially decreased the cell wall and organelle fraction, while no significance was observed for the soluble fraction. Whereas, Si significantly increased cell Cd wall fraction. Nevertheless, compared with Cd stress, individual Si and Se additions had no significant impact on subcellular distribution as compared to Cd-exposed plants. However, upon co-application of Si + Se, a significant (*p* < 0.05) overall reduction in the soluble Cd fraction was observed as compared to other Cd fractions. In roots, Se as compared to Si supplies remarkably reduced total Cd and Cd fractions (Figure 1e,f). The effect was profound (*p* < 0.05) when Se was applied with Si; the cell wall-bound Cd and organelle Cd fractions were reduced by 2.3– and 1.3– folds, respectively, as compared to control Cd-fed plants.

#### 2.2.3. Cd Tolerance and Uptake Characteristics

Results from the Tolerance Index indicated that Cd-induced significant toxic impacts on wheat growth and dry biomass production (Figure 2a), which were uplifted upon Se and Si application. Both Se and Si significantly improved plant growth and biomass (Table 1) and thereby increased Cd tolerance in wheat.

Bio-concentration of Cd (roots/medium) was significantly affected by both Se and Si treatments. Significantly lower Cd was concentrated in roots upon Se and Se + Si treatments. Compared to Se, Si supplementation even facilitated medium-to-root translocation of Cd and Cd retention in roots, which is correlated with decreased total Cd and soluble Cd fractions in young and older leaves by Si (Figure 2b). Whereas, comparatively less bio-concentration of Cd in root was observed upon Se treatment, which is correlated to the increased total Cd concentration in shoot (Figure 1a,c). Cd translocation from roots to shoots in young leaves was significantly increased upon Se application, which is correlated to increased total Cd uptake in shoots (Figure 1a,c) and decreased Cd concentration in roots (Figure 1e) as compared to Si-treated plants. In older leaves, the TF values were not significantly affected by Si supply, but the addition of Se accelerated Cd translocation from the root to the older leaves. Cd enrichment results indicated a similar impact of Si and Se on decreasing Cd concentration in the shoot with respect to the medium. The highest significant decrease in Cd enrichment was observed in Se + Si treatment, and more profound results were observed in older leaves (2.1– folds) as compared to young leaves (1.2– folds).

### 2.3. Se-Si Interaction on Wheat Elemental and Nutritional Composition under Cd Stress

The uptake of macro- and micronutrients was analyzed in plant tissues (root and shoot) to investigate the wheat national composition and isonomic interactions for plant uptake. The results from elemental uptake indicated that the combined treatment Si-Se-Cd differentially interacts with plant macro- and micronutrients for uptake (Appendix A).

The correlation analysis indicated a strong negative relationship between Se and Si and Cd accumulation in the root and shoot, respectively (Figure 3a,b). In roots, the highest significant negative correlation was found with Si, followed by Se for Cd uptake and translocation to the shoot, whereas in the shoot, Se showed a strong negative correlation towards Cd uptake and translocation. Additionally, a weak but positive correlation interaction between Si and Se for relative uptake was observed, which had not been reported earlier. A strong negative correlation was also observed for S, followed by K > Zn > Cu > P in roots upon Si and Se application. Whereas, in the shoot, a strong negative relation was observed for K > P > Mn > Zn, whereas S showed positive interaction for Cd, Se, and Si uptake.

### 2.4. Se-Si Reduces Cd-Induced Toxicity and Regulates Antioxidant Enzyme Activities

Exposure to Cd significantly induced the production of reactive oxygen species (ROS), lipid peroxidation, and membrane damage.

Figure 4a indicates the extent of lipid peroxidation upon Cd exposure expressed as MDA. The results indicated significantly (*p* < 0.05) high induction of lipid peroxidation in roots (7.8–folds) as compared to shoots (3.1–folds) under Cd exposure (10 mM). Application of Si and Se both reduced MDA content; however, compared to Si, Se supply significantly reduced lipid peroxidation in root and shoot by 2.4– and 2.1–folds, respectively, in Cd-stressed plants. A similar trend was observed for H_2_O_2_. The H_2_O_2_ represents the overall oxidative damage incurred by plants under toxicity (Figure 4b). In the present study, exposure to Cd resulted in significant production of 2.2– and 1.5–folds in root and shoot, respectively. Which was equally uplifted by both Si and Se applications. However, Si + Se treatment significantly reduced H_2_O_2_ production in root and shoot by 1.4– and 1.3–folds, respectively. Membrane damage and electrolyte leakage were calculated and expressed as membrane stability (MS) and integrity (MI) indices (Figure 4c,d). It was observed that membrane damage and electrolyte leakage were more profound in the root (3.6– and 15.9–folds) as compared to the shoot (3.4– and 8.6–folds). The highest significant membrane damage in root/shoot was recorded in plants exposed to 10 µM Cd Conc. Si/Se supplementation equally and significantly (1.29–folds) reduced the membrane damage in plants under Cd stress. While the highest significant decrease in membrane damage (1.9–folds) and electrolyte leakage (2.4–folds) in root/shoot was observed with a synergistic effect of Si + Se under Cd exposure. Roots were also analyzed for ROS and Cd localization through the Invitrogen technique. The images obtained are presented in Figure 4e,f, respectively. Results indicate less ROS in control roots. In Cd-treated plants, strong ROS and Cd signals were observed as compared to control and other treatments. With regard to Se and Si treatment, it can clearly be observed that Se decreased ROS production while increased Cd localization in the root was observed in the case of Si treatment, resulting in more production of ROS. In the case of combined treatment, more Cd was localized in the root while less ROS was produced, indicating activation of the antioxidant response.

Figure 5 represents the protein and antioxidant response in wheat upon Si and/or Se treatments under Cd stress. Results indicated that Cd had significantly decreased the total protein content in root and shoot by 1.3– and 1.2– fold, respectively, as compared to non-Cd-stressed plants. Treatments with Se + Si significantly (*p* < 0.05) increased the protein content of root/shoot in the absence (2.2–/1.3– folds) and presence (2.8–/1.5– folds) of Cd, as compared to individual Se/Si treatments. To investigate the activation of the antioxidant system, Catalase (CAT), Ascorbate peroxidas (APX), Glutathione (GSH), Glutathione reductase (GR) Glutathione disulfide (GSSG) were measured. As shown in Figure 5a–c CAT, APX, and GR showed differential responses to Cd stress in plant tissues under Si and/or Se treatments. GR, CAT, and APX activities were more profound in the shoot than in the root. Upon Cd exposure, a significant decrease in CAT (1.2–, 0.9– folds) and APX (1.3–, 1.2– folds) activities was observed in plant tissues (root and shoot, respectively). Whereas, GR activity was unaffected by Cd stress. Applying both Se and Se + Si significantly increased tissue APX and GR activities. The APX activity in the root was significantly enhanced by 1.8– and 2.1– folds under the Se and Se + Si treatments, which were greater than the treatment containing only Si. As shown in the figure, Si and/or Se had no effects on the root/shoot CAT activities compared to the control treatment (without Cd). The decrease in CAT activity (1.2– folds) upon Cd exposure was ameliorated upon Si/Se/Si + Se treatments (1.3– folds) with no significant difference (*p* < 0.05). Similar trends were also observed in tissue GR activity with the Se supply and treatments containing combined applications of Si and Se (Figure 4d). A significant increase in root GR activity was observed with Se treatment (1.9– folds) and combined treatment (2.8– folds). In shoot, the effect was even doubled; the Se + Si treatment increased the GR activity by 4.4– folds, thereby reducing the toxic impacts of Cd and improving plant biomass and physiological growth parameters (Table 1). Interestingly, we found that with the application of Se, a significant increase in GSH (2– folds) content was observed, which led to the increased antioxidant capacity of wheat to tolerate Cd toxicity as indicated by the tolerance index (Figure 2b). Meanwhile, a substantial decrease in GSSG content was also observed, which also complements the decrease in total S content (Figure 5e,f and Appendix A). This decrease in total S was more profound with the combined application of Se and Si, as confirmed by Pearson’s correlation (Figure 3).

## 3. Discussion

### 3.1. Cd Uptake Characteristics

Cd exposure significantly reduced plant biomass and growth attributes. All growth parameters, including biomass, tillering, shoot length, chlorophyll, and photosynthesis, showed Cd toxicity and reduced production. The toxic impacts of Cd on plant physiological growth increased upon the application of Se and Si treatments. The comparatively highest significant (*p* < 0.05) effect was noticed upon co-application of Se + Si treatment followed by Se treatment, respectively (Table 1). The increase in plant biomass upon co-application of Se + Si may be attributed to Se-induced plant physiological and enzymatic regulation against Cd-induced oxidative and metabolic damages [20] and plant morphological adaptations to reduce Cd uptake and structural damages [19]. Abdalla et al. [15] also revealed a positive impact of Se on water-soluble sugars under S-deficient conditions in Lettuce. The study reported an increase in glucose and fructose (70.84 ± 1.1 and 115.0 ± 2.1 mg g^−1^ DM; and 109.4 ± 2.1 and 161.1 ± 1.0 mg g^−1^ DM, respectively) under 1.3 and 3.8 μM Se levels. In a recent study, Se and Si were applied individually, and results indicated a significant reduction of Cd toxicity by down-regulating influx transporter (TaNramp5) and up-regulating efflux transporter (TaTM20 and TaHMA3) genes. Se/Si treatments also increased grain yield, antioxidant enzyme activities, and reduced MDA in wheat tissues [21]. Past studies reported that Se applications were more effective than Si on plant physiological growth under Cd stress [16,20]. In contrast, our study indicated significant *p* < 0.05 plant physiological growth with Si and Si + Se treatments under Cd stress. Results from the Tolerance Index indicated that Cd-induced significant toxic impacts on wheat growth and dry biomass production (Figure 2a), which were uplifted upon Se and Si application. Both Se and Si significantly improved plant growth and biomass and increased Cd tolerance.

Wheat root and shoot Cd concentrations were dramatically reduced by both Si and Se. Compared to Se, plants treated with 0.2 mM Si had significantly lower shoot and root Cd concentrations. A synergistic reaction in Cd reduction was detected in the shoot and root, respectively, when Se and Si were administered combined, and this substantial reduction was higher. Plants generally employ two strategies to resist heavy metal toxicity, including Cd tolerance and avoidance [22]. The data indicated dual strategies induced by Si and Se for Cd removal and stress management by wheat plants. Si-induced Cd removal (Figure 1a,c,e). Whereas Se induces Cd tolerance by regulating the antioxidant defense system (Figure 4). Moreover, Cd was accumulated mostly in the roots, and less Cd was translocated to the shoot. In the shoot, more Cd was accumulated in older leaves than in younger leaves. Nevertheless, compared with Cd-fed plants, individual Se addition significantly reduced total Cd in booth root and shoot but had no significant impact on subcellular distribution percentage. This could be explained by the fact that Se supply alters the expression of transporter genes involved in cadmium uptake and translocation in winter wheat. It is reported that the expression of TaNramp5-a, TaNramp5-b, and TaHMA2 Cd-transported genes was significantly downregulated by increasing Se supply [20]. However, upon co-application of Se + Si, a significant (*p* < 0.05) overall reduction in soluble Cd fraction was observed in both root and shoot (Figure 1e,f), reflecting that Si was the main element responsible for modulating Cd fractionation. Si induces structural changes in the root cell wall and endodermis through the deposition of lignin and suberin [18]. In addition, Si also induces oxalate exudation, which also contributes to Cd removal and the alleviation of cadmium toxicity in wheat [19]. A significant decrease in Cd concentration in the cell wall, soluble fraction, and cell organelles in the root and shoot was also observed upon increasing Se supply.

Bioconcentration of Cd (roots/medium) was significantly affected by both Se and Si treatments. Significantly lower Cd was concentrated in roots upon Se and Se + Si treatments. Compared to Se, Si supplementation even facilitated medium-to-root translocation of Cd and Cd retention in roots, which is correlated with decreased total Cd and soluble Cd fractions in young and older leaves by Si (Figure 2b). Cd translocation from roots to shoots in young leaves was significantly increased upon Se application, which is correlated to increased total Cd uptake in shoots (Figure 1a,c) and decreased Cd concentration in roots (Figure 1e) as compared to Si-treated plants. The increased translocation upon Se treatment may also be attributed to the up-regulation of the plant antioxidant defense system for the detoxification of Cd. Se unregulated the production of GR and APX involved scavenging free radicals generated as a result of Cd toxicity. Se treatment increases the levels of glutathione (GSH) and phytochelatins (PCs), as well as the expression of GSH and PC biosynthetic genes involved in Cd complexation and detoxification in tomatoes [23]. Cd enrichment results indicated a similar impact of Si and Se on decreasing Cd concentration in the shoot with respect to the medium.

### 3.2. Nutrient Composition and Isonomic Interactions

Results indicated that Cd had significantly decreased the total protein content and impacted plant macro- and micronutrient uptake. The correlation analysis indicated a strong negative relationship between Si and Se and Cd accumulation in the root and shoot, respectively. Si is well reported to induce structural and morphological changes in the wheat root that lock Cd in the root cell wall and limit Cd uptake in the shoot [24]. On the other hand, Se-up regulates several secondary metabolites, including phenolics, flavonoids, and their derivatives, as well as glucosinolates, which reduce Cd uptake and toxicity [25]. Additionally, a weak but positive correlation interaction between Si and Se for relative uptake was observed, which had not been reported earlier. Hence, the co-application of Si and Se demonstrated a positive correlation for each other’s uptake while accelerating Cd removal synergistically. In addition, a strong negative correlation was also observed for S, followed by K > Zn > Cu > Mn, and P in roots upon Si and Se application. Whereas, in the shoot, S showed positive interaction for Cd, Se, and Si uptake. The reason could be that an increased Se supply through foliar application could induce the development of the group 1 isoform, Sultr1; 1, and the group 2 isoform, Sultr2; 1, which promotes S accumulation in plants [26]. Saeed et al. [27] recently investigated the interplay of Se and S for nutrient uptake in spinach and reported that Se positively correlates to the mineral (except Mn) uptake in roots grown under S-limiting conditions. It also increased the concentration of organic acids (malic acid and citric acid) under S-deficient conditions, which are well-known chelating agents for Cd and other heavy metals [28].

### 3.3. Activation of Antioxidant Resistance

Exposure to Cd significantly induced lipid peroxidation, the production of ROS and H_2_O_2_, membrane damage, and electrolyte leakage. Si/Se supplementation equally and significantly reduced these oxidative and structural damages concurred under Cd stress. A synergistic effect of Si + Se was observed in reducing Cd toxicity, as evidenced by the root localization of ROS and Cd (Figure 4d,f). This may be attributed to the upregulation of the plant antioxidant defense system through Se supply and reduced Cd translocation through stabilizing Cd in root cells through structural changes in the root cell wall and endodermis [18]. Furthermore, activating the antioxidant enzyme system, including CAT, APX, and GR, reduced the Cd toxicity in Se and Se + Si supplemented plants. CAT and APX scavenge excess H_2_O_2_ by catalyzing it into water and divalent oxygen [24]. Whereas, GR is an essential and crucial enzyme that scavenges ROS by maintaining a high concentration of GSH that ultimately catalyzes the conversion of H_2_O_2_ to water 24]. The application of both Se and Se + Si significantly increased tissue APX and GR activities, thereby reducing the toxic impacts of Cd and improving plant biomass and physiological growth parameters (Table 1). Interestingly, we found that with the application of Se, a significant increase in GSH content was observed, which led to the increased antioxidant capacity of wheat to tolerate Cd toxicity as indicated by the tolerance index (Figure 2b). Meanwhile, a substantial decrease in GSSG content was also observed, which also complements the decrease in total S content (Figure 3 and Figure 5e,f). And this decrease in total S was more profound with the combined application of Se and Si, as confirmed by Pearson’s correlation (Figure 3) and possible competition for the relative uptake of Si, Se, and S [24,28].

### 3.4. Mechanistic Insight in to the Combined Effect of Se and Si

The data indicated dual strategies induced by Si and Se for Cd removal and stress management by wheat plants. Si-induced Cd removal (avoidance mechanism) (Figure 1a,c,e). Whereas Se induces Cd tolerance by regulating the antioxidant defense system (tolerance mechanism) (Figure 4). The co-application of both Se and Si showed the highest significant (*p* < 0.05) Cd removal and toxicity tolerance as compared to a single application, making wheat resistant to Cd uptake. We therefore proposed a new mechanism for Se interplay with S in reducing Cd toxicity and limiting Cd uptake and translocation through Si-induced modulation of Cd subcellular distribution (Figure 6). Se was negatively correlated with S uptake in roots; as a result, a decline in total elemental S was observed. However, an increase in reduced GSH and GR activity was observed (Figure 3 and Figure 5d,e, Appendix A). This depicts the internal homeostasis and ability of wheat to compensate for S deficiency by replacing it with Se.

S is the major contributor to different enzymes and protein biosynthesis [15]. The exogenous application of GSH treatment could reduce Cd toxicity in wheat by upregulating GSH synthesis gene expression or by downregulating the TaNramp1, TaNramp5, TaHMA2, TaHMA3, TaLCT1, and TaIRT2 Cd transporter genes in roots [24]. Moreover it is elucidated in a recent study that, Se-induced Cd tolerance involves melatonin which regulates Cd detoxification [29]. Here in our study, we showed that, under S deficiency, Se could replace S in S-containing defense compounds. The decrease in S content could be explained by the presence of silicate competition and S substitution, as selenate was applied via the foliar mode of application and Si was applied via the root. Se easily penetrates the tissues of leaves via mesophyll cells, most likely through S transporters [25], whereas Si is taken up by OsLsi1 and OsLsi2. A recent study has highlighted the Si-induced regulation of transcriptomic and metabolic homeostasis in rice under S-limiting conditions. The expression of the S transporter OsSULTR tended to decrease with Si supply [30]. Moreover, the supplementation of Si reduced Cd concentration in wheat by 67.45% (root) and 70.34% (shoot), and maintained ionic homeostasis through regulating important transporters, such as Lsi, ZIP, Nramp5 and HIPP [31]. Taken together, it is evident that the interaction of Se and S and the enhancement of GSH were the causes of decreasing elemental S and uplifting the Cd resistance in wheat.

## 4. Materials and Methods

### 4.1. Plant Culture and Experimental Design

Wheat (*Triticum aestivum* cv. JB Asano) seeds were surface sterilized with sodium hypochlorite solution (0.5% *v*/*v*) for 15 min, then washed 3 times with deionized (DI) water and soaked for 48 h. The seeds were then germinated in a sandwich setup and placed in the dark for 5 days at 25 °C, followed by 3 days in the greenhouse after germination. The uniform-sized seedlings were then transplanted into a black plastic pot (5 L) containing 5 L of a 0.2 × nutrient solution [25]. The composition of the basal nutrient solution included macronutrients: 2 mM Ca(NO_3_)^2^, 1 mM K_2_SO_4_, 0.5 mM MgSO_4_, 2 mM CaCl_2_, 0.2 mM KH_2_PO_4_, 0.2 mM Fe-EDTA, and micronutrients: 10 µM H_3_BO_3_, 2 µM MnSO_4_, 0.5 µM ZnSO_4_. 7H_2_O, 0.3 µM CuSO_4_, 0.01 µM (NH_4_)Mo_7_O_24_ (composition of nutrient solution is given in Appendix A). Following the next few weeks (7 days), the plants were applied with 0.5 × and full-strength nutrient solutions, respectively.

The experimental design included 7 treatments with 4 replicates in a completely randomized block design (RCBD); (1) Control; (2) Se: (1.5 µM Na_2_SeO_3_); (3) Se + Si (2 mM Na_2_SiO_3_) (4) Cd (10 μM CdCl_2_); (5) Cd + Se; (6) Cd + Si; (7) Cd + Si + Se. The concentrations of Cd [1], Si [20], and Se [11] were chosen based on preceding studies. Treatments without Si and Se were supplemented with a pinch (~2 mg/L) of NaCl to adjust extra Na+ ions in Si and Se treatments as sodium salts. The pH of nutrient solutions was maintained at 6 ± 0.2 using 0.5 M NaOH and HCl solutions. Plants were grown under standard greenhouse conditions (18 °C/14 °C; 14 h/8 h day-night regime), and the nutrient solution was changed weekly. Results of Visual MINTEQ (version 3.1, Swedish University of Agricultural Sciences, Sweden) revealed that Si does not interfere with chemical speciation or Cd activity in the studied experimental conditions (Appendix A). However, the addition of Si can reduce the solution pH (ΔpH = 0.14) (Appendix A).

### 4.2. Determination of Photosynthetic Attributes and Physiological Parameters

Before harvesting, the chlorophyll content was recorded using SPAD (SPAD-502, Konica Minolta Inc, Marunouchi, Japan), and other photosynthetic attributes, including the rate of photosynthesis (Ps) and transpiration rate (Tr), were monitored from 10:00 a.m. to 11:00 a.m. using the portable photosynthesis system (LI-6400XT; Li-COR Biosciences Inc., Lincoln, NE, USA). At the anthesis stage, the plants were harvested, and the roots were soaked in a 0.5% HCl solution to remove adsorbed elements, followed by three times of rinsing in DI water. After washing, plant samples were dried, dissected into root and shoot, weighed, and measured for root/shoot length and the number of tillers. Fresh samples of root and shoot (15 g each) were immediately frozen via liquid nitrogen and stored at −80 °C in the ultra-low temperature refrigerator for biochemical and antioxidant analysis. The remaining samples of root and shoot were oven-dried (65 °C) until a constant weight was achieved. The dried samples were then weighed and ground to a fine powder for elemental analysis.

### 4.3. Subcellular Distribution of Cd

The determination of Cd in different components (cell wall (CW), soluble fraction (SF), and organelle fractions (OF) was carried out following the methods described by Wu et al. [19] with some modifications. Briefly, fresh samples of the root, young leaves, and old leaves (1 g) were ground with extraction solution (0.25 M sucrose, 50 mM Tris-HCl buffer (pH 7.5), and 1 mM dithioerythritol) in chill conditions. The homogenate was then filtered, and the obtained pallet was rewashed with the extractant. For cell wall extraction, the filtrate was centrifuged (300× *g* for 30 s), and the obtained pallet, together with the washed pallet, was considered a cell wall component. The filtrate obtained was centrifuged again at 20,000× *g* for 45 min using a Micro-centrifuge (Thermo Scientific™ Heraeus, Hanau, Germany), and the obtained pallet (OF) and filtrate (SF). For Cd determination, the soluble fraction was diluted and used directly, whereas SW and OF components were wet digested with 10 mL HNO_3_ (65%) following laboratory protocol using a microwave oven digestion system (Mars 6, CEM Corporation, Matthews, NC, USA) and diluted before Cd was determined through inductively coupled plasma mass spectrometry (ICP-MS; Agilent 7700, Agilent Technologies Inc., Böblingen, Germany).

### 4.4. Elemental Analysis and Cd Uptake Characteristics

The oven-tried samples were ground to a fine powder through a ball mill (MM 200, Retsch GmbH, Haan, Germany) and digested (0.5–0.7 g) with 10 mL of concentrated HNO_3_ (65%) at 190 °C for 45 min in a microwave digester (Mars 6, CEM Corporation, Matthews, NC, USA) following protocols described by [19]. The digests were diluted to 50 mL with double DI water and stored for lateral analysis. The concentrations of Se, Cd, Zn, Cu, and Mn were determined through inductively coupled plasma mass spectroscopy (ICP-MS; Agilent 7700, Agilent Technologies Inc., Böblingen, Germany), and concentrations of Ca, Mg, P, S, Fe, and Na were determined through Inductively Coupled Plasma Optical Emission Spectroscopy (ICP-OES) as described by Wu et al. [24]. For making standard curves, a multi-element ICP-standard solution (ROTI^®^STAR, Roth, Germany) was used. The Cd uptake and translocation characteristics were calculated using the following Formulas (1)–(4);
Cd uptake (Cd conc. per plant biomass) = Cd_[root]_ * DW_root_ + Cd_[shoot]_ * DW_shoot_(1)
Translocation Factor (TF) = Cd_[shoot]_/Cd_[root]_(2)
Bioconcentration Factor (BCF) = Cd_[root]_/Cd_[medium]_(3)
Enrichment Factor (EF) = Cd_[shoot]_/Cd_[Medium]_(4)

### 4.5. Localization of Cd and ROS

To determine the impact of Cd on ROS production and to localize the distribution of Cd in wheat roots, 2-week-old seedlings of *T. aestivum* were exposed to experimental treatments as discussed in Section 4.1 for two weeks. The Cd localization was visualized using the LeadmiumTM Green AM dye (Cd-specific probe) (code: A10024, Thermo Helios Gamma, 9423 UVG 1000E, Leicester, UK). Briefly, 5 µg of Green AM dye was added to 50 μL of DMSO to make a stock solution. Afterwards, 5 µL of stock was added to 450 uL of 0.85% NaCl to make a working solution. For staining and microcopy, 10 primary root tips per treatment were immersed in the working solution for 3 h in the dark at 25 °C, followed by three times washing with 0.85% NaCl. Micrographs were taken using the Confocal-Zeiss LSM 710 laser scanning system (Carl Zeiss Iberia, S.L., Madrid, Spain). [32].

ROS localization in root tissue was observed through invitrogen ROS technology, as described by Abid et al. [32]. The root tips (2 cm) were excised and immersed in 1 mL (10 µM) of DCF (2′, 7′-dichlorofluorescein) and placed in the dark for 1 h. For positive control, the root tips of the control plant were treated with 1% H_2_O_2_ before immersing in dye. After 1 h, the tips were washed with DI water to remove the excessive dye and kept in 4-(2-125 hydroxyethyl)-1-piperazineethanesulfonic acid (HEPS) buffer. The sample roots were then examined under a confocal laser scanning microscope (Confocal-Zeiss LSM 710 laser scanning system, Carl Zeiss Iberia, S.L., Madrid, Spain).

### 4.6. Estimation of ROS and Antioxidative Response

The membrane integrity index (MS) was determined through staining with evince blue [32]. Membrane integrity was determined as a measure of absorbed dye onto the damaged cell membranes. Briefly, 1 g of fresh roots was stained with a 0.025% (*w*/*v*) dye solution (evince blue in 100 mM CaCl_2_; pH 5.6) for 10 min, followed by destaining with 100 mM CaCl_2_. After washing, the root samples were homogenized with a 1% solution of (*w*/*v*) sodium dodecyl sulfate and centrifuged using a micro-centrifuge (Thermo Scientific™ Heraeus) at 15,000 rpm for 20 min. The supernatant thus obtained was analyzed through a spectrophotometer at 600 nm wavelength. To further check the stability of the membrane, the electrolyte leakage was estimated from plant tissue as a result of membrane rupture and destabilization under oxidative stress [32]. For this purpose, 1 mg of Fresh root shoot biomass was added to a glass vial containing 5 mL of DI water. Two sets of samples were prepared and placed in a water bath (Water Bath, Medingen GmbH, Ottendorf-Okrilla, Germany) to heat up to 40 °C and 100 °C, respectively. The electrical conductivities were recorded for each set, and the Membrane stability (MI) index was determined as MI = (C40/C100) * 100 [25].

For hydrogen peroxide (H_2_O_2_) estimation, 0.2 g of root/shoot tissue was homogenized manually using a mortar and pestle with 0.1% trichloroacetic acid (TCA), followed by centrifugation at 4 °C at 12,000× *g* for 20 min using a micro-centrifuge (Thermo Scientific™ Heraeus, Hanau, Germany). The supernatant (0.5 mL) was then mixed with 0.5 mL of 0.1 M k-p buffer (pH 7.8) and 1 mL of 1 M KI and kept in the dark. After 1 h, the absorbance of samples was recorded at 390 nm, and H_2_O_2_ was estimated using a standard curve [33]. For MDA content, 0.5 g of fresh root/shoot tissue, dried in liquid nitrogen, was homogenized with a 5% TCA solution. The sample was then centrifuged at 12,000× *g* for 20 min at 4 °C, and the supernatant was used for the estimation of MDA. The MDA content was estimated following the method described by Abid et al. [32]. Briefly, 0.5 mL of sample extract was mixed with 1.5 mL of 0.5% thiobarbituric acid (TBA) in 5 mL glass vials and heated to 95 °C for 30 min in a water bath, followed by quick cooking. After cooling the reaction mixer, the absorbance of the orange/yellow color developed was taken at 532 nm and 600 nm, respectively, using the Helios Omega UV-Vis spectrophotometer (Thermo Helios Gamma, 9423 UVG 1000E, Leicester, UK).

### 4.7. Determination of Antioxidant Enzyme Activity

For enzyme extraction, 0.5 g of fresh root/shoot tissue was dry-frozen with liquid nitrogen, followed by homogenization and extraction with 1.5 mL of standard extraction buffer using a mortar and pestle. The homogenate was then transferred to 2 mL eppendorf tubes and set for centrifugation at 12,000× *g* for 20 min at 4 °C. The supernatants were then collected and transferred to −80 °C for estimation of enzyme activities. Glutathione reductase (GR; EC: 1.6.4.2) activity was measured based on the reduction of oxidized glutathione (GSSG) to reduced glutathione (GSH) (at 560 nm wavelength) [33]. GSH and GSSG were also measured spectrophotometrically at 412 nm using a Helios Omega UV-Vis spectrophotometer (Thermo Fisher Scientific, Waltham, MA, USA).

Ascorbate Peroxidase (APX; EC: 1.11.1.11) governing the oxidation of ascorbic acid to monodehydroascorbic acid (MDASA) was monitored by recording the decrease in absorbance at 290 nm [34]. Catalase (CAT; EC: 1.11.1.6) activity was estimated by analyzing the concentration of H_2_O_2_ consumed; the absorbance of samples was recorded at 240 nm [35]. Protein estimation was performed using the Bradford method [36]. The intensity of the blue color was recorded at 595 nm through a Helios Omega UV-Vis spectrophotometer (Thermo Helios Gamma, 9423 UVG 1000E, Leicester, UK).

### 4.8. Statistical Analysis

The data presented in this study were statistically analyzed by a one-way analysis of variance (ANOVA) test using SPSS software (SPSS Inc., 16.0). The difference among treatments was further determined through Duncan’s multiple range test. The level of significance for all statistical analyses was kept at 0.05. To analyze the interaction among Si-Se-Cd, Pearson’s correlation test was performed using SPSS software (SPSS Inc., 16.0).

## 5. Conclusions

In this study, the uptake and distribution of Cd in wheat were investigated with individual and combined treatments of Se and Si. The hydroponic experiment allowed us to simplify the complexity of the system and determine the dominant influencing factors and mechanisms opted for for the safe removal of Cd in wheat. This study concluded and confirmed our hypothesis of achieving Cd resistance in wheat upon co-application of Se and Si. Briefly, the Si supply achieved better Cd reduction and translocation by facilitating Cd immobilization in hydroponic solution by forming precipitates (as CdSO4°), increasing Cd accumulation in root cell wall fraction, and decreasing overall Cd uptake through a regulating avoidance mechanism at the root interface. Whereas, Se strengthens the plant tolerance mechanism to decrease Cd toxicity and uptake by roots by regulating the antioxidant system and thereby improving plant growth under Cd stress. A decline in total S upon Se + Si application while increasing the reduced GSH content was the main mechanism for improving Cd tolerance in wheat. Taken together, our results suggested that Se decreased the elemental S because of competition for root uptake while increasing the reduced S-GSH synthesis, which alleviated the Cd toxicity in wheat by suppressing the reactive oxygen species and Cd uptake and translocation in wheat plants. Moreover, this study provided a piece of baseline information and theoretical background on the co-application of Si with Se to promote wheat growth under Cd stress conditions.

## Figures and Tables

**Figure 1 ijms-25-00387-f001:**
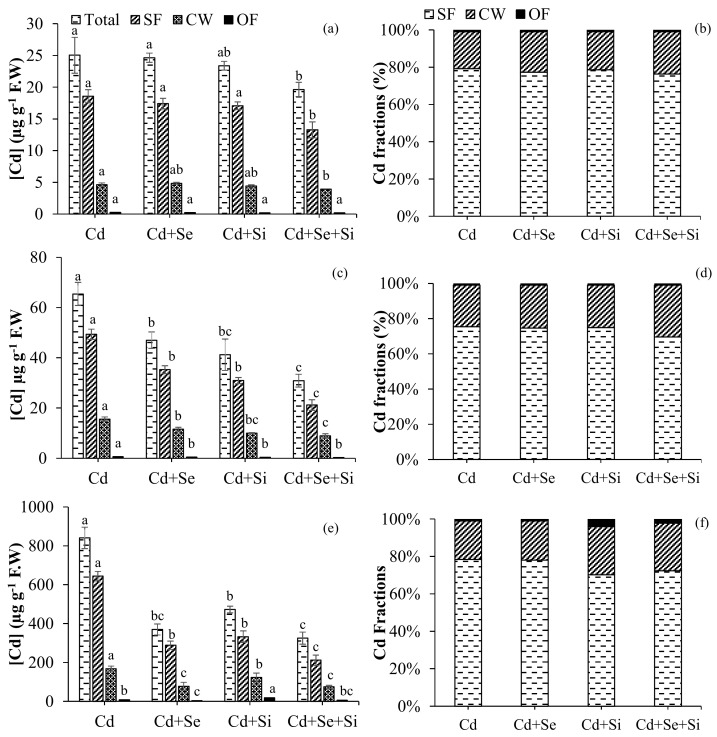
Effects of Se and Si on total and subcellular Cd fractions in different plant tissues: young leaves (**a**,**b**); older leaves (**c**,**d**); and roots (**e**,**f**) of wheat under Cd exposure (10 µM Cd). SF, CW, and OF correspond to soluble fractions, cell wall fractions, and organelle fractions, respectively. The bars are means ± standard error of four replicates. The data are analyzed for statistical significance (*p* < 0.05) using a 3-way analysis of variance (ANOVA) followed by the least significant difference (LSD) test. Different letters above bar indicate a statistical difference at *p* < 0.05.

**Figure 2 ijms-25-00387-f002:**
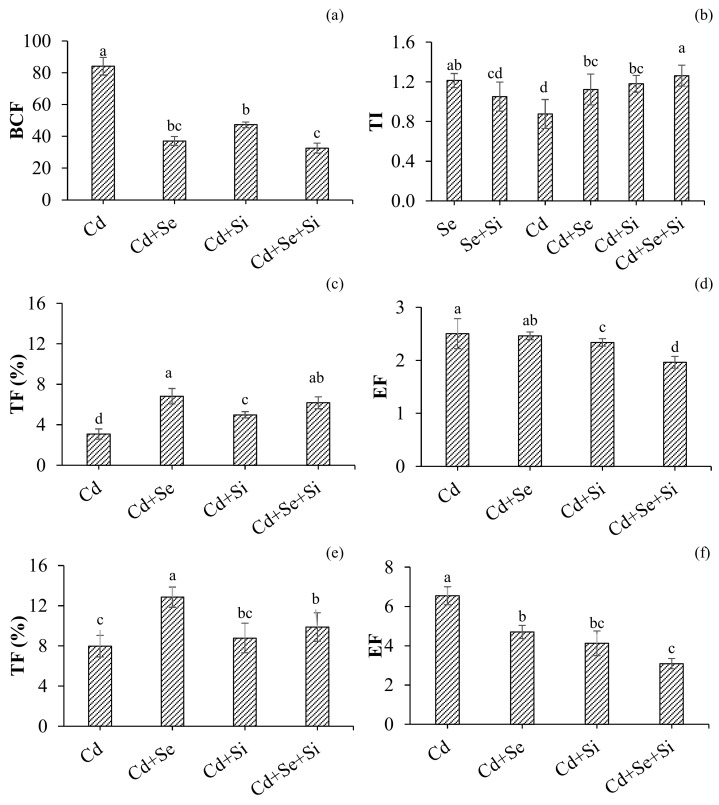
Se and Si supply affect Cd bioconcentration (**a**), tolerance index (TI) (**b**), and Cd translocation characteristics (**c**–**f**) in wheat under Cd stress conditions. The bars are means ± standard error of four replicates. Different letters indicate a statistical difference at *p* < 0.05. BCF (bioconcentration factor) = [Metal]root/[Metal]soil. TI (tolerance index) = Biomass of the Cd-stressed plant/biomass of the control plant. TF (translocation factor) = [Metal]shoot/[Metal]root. EF (enrichment factor) = [Metal]shoot/[Metal]medium.

**Figure 3 ijms-25-00387-f003:**
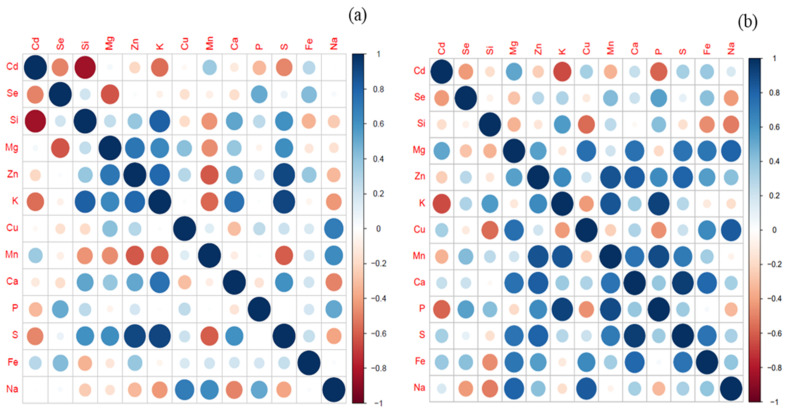
Pearson’s correlation coefficients (r) between different metals and nutrients for possible interactions in root (**a**) and shoot (**b**), respectively, for relative uptake by wheat plants. Significance is calculated at *p* < 0.05. The figure can be perfectly understood if it is viewed in color or in the online version.

**Figure 4 ijms-25-00387-f004:**
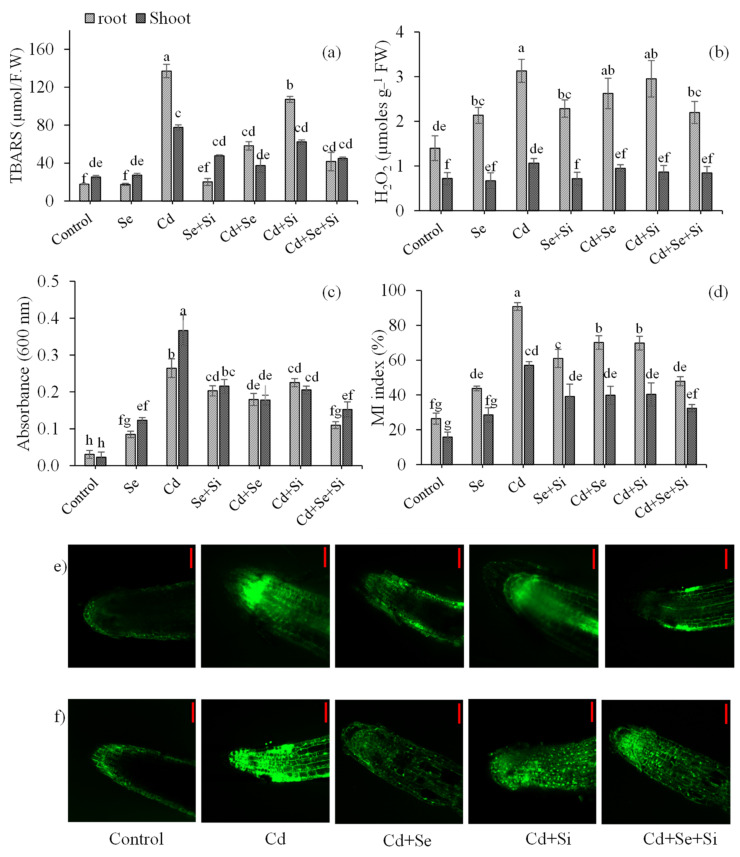
Effect of Si and Se applications on oxidative stress markers. MDA (**a**); H_2_O_2_ (**b**); membrane injury (MI) measured by Evans blue uptake (**c**); and Membrane and stability index measured by measuring electrolyte leakage (**d**) in root and shoots, respectively. Cd (**e**) and ROS localization (**f**) in root tips after growing for 2 weeks on a nutrient solution supplemented with 10 µmol Cd, 0.2 mmol Si, and 1.5 µmol Se, respectively. The bars are means ± standard error of four replicates. Different letters indicate a statistical difference at *p* < 0.05.

**Figure 5 ijms-25-00387-f005:**
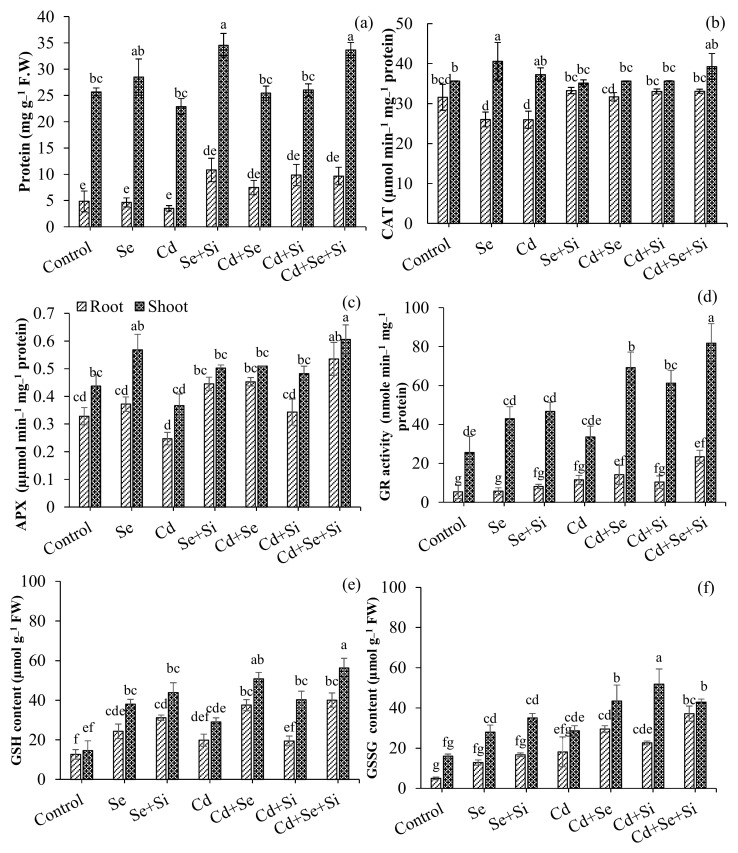
Effect of Silicon Si and Se application on total protein content (**a**), catalase (CAT) content (**b**), ascorbate peroxidase (APX) content (**c**), glutathione reductase (GR) content (**d**), reduced glutathione (GSH) content (**e**), and oxidized glutathione (GSSG) content (**f**) in roots and shoots of wheat under Cd stress conditions. The bars are means ± standard error of four replicates. Different letters indicate a statistical difference at *p* < 0.05.

**Figure 6 ijms-25-00387-f006:**
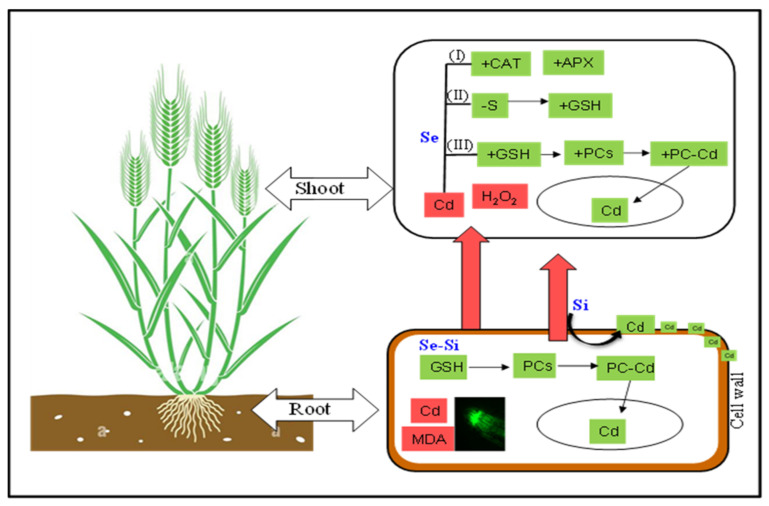
The proposed mechanism of Se-Si interplay to combat Cd toxicity under Cd stress conditions is through different mechanisms, including; I- increase in catalase (CAT) and ascorbate peroxidase (APX) activity; II- decrease in elemental S but increase in GSH content indicating substitution of S by Se; and III- regulation of glutathione (GSH) metabolism, phytochelates (PC) synthesis and Cd detoxification.

**Table 1 ijms-25-00387-t001:** Effect of Se and Si on plant growth and physiological parameters under Cd stress. The data presented is the mean of 4 replicates± standard error. Data are analyzed using One-way analysis of variance (ANOVA), followed by the least significant difference (LSD). Different letters indicate significant differences (*p* < 0.05).

Parameters ►	Dry Biomass	Shoot Length	Tillers	Chlorophyll	Photosynthesis	Transpiration
Treatments ▾	(g)	(cm)	(No.)	Units	(µmol CO_2_ m^−2^ s^−1^)	(µmoles H_2_O_2_ m^−2^ s^−1^)
Control	^c^ 115.6 ± 5.1	^b^ 79.75 ± 3.6	^e^ 26.8 ± 0.5	^c^ 47.8 ± 2.3	^b^ 18.2 ± 1.9	^b^ 3.4 ± 0.1
Se	^a^ 140.3 ± 8.2	^b^ 77.25 ± 7.9	^bcd^ 31.3 ± 1.0	^a^ 50.8 ± 2.4	^b^ 19.7 ± 2.5	^a^ 3.7 ± 0.3
Cd	^d^ 101.3 ± 9.7	^d^ 67.8 ± 5.5	^de^ 27.3 ± 1.4	^d^ 40.9 ± 1.7	^c^ 15.7 ± 1.4	^c^ 2.9 ± 0.6
Se + Si	^bc^ 121.5 ± 8.8	^a^ 87.5 ± 4.4	^cde^ 31 ± 1.9	^a^ 50.5 ± 1.6	^a^ 22.3 ± 0.7	^a^ 3.9 ± 0.1
Cd + Se	^abc^ 129.8 ± 4.1	^cd^ 71.5 ± 7.9	^abc^ 33.8 ± 1.9	^ab^ 49.8 ± 1.0	^b^ 18.2 ± 1.7	^a^ 3.8 ± 0.5
Cd + Si	^ab^ 136.5 ± 8.7	^ab^ 80.25 ± 5.6	^ab^ 34.3 ± 0.9	^bc^ 47.7 ± 1.0	^b^ 18.7 ± 1.6	^a^ 4.1 ± 0.3
Cd + Se + Si	^a^ 146 ± 7.4	^ab^ 80.25 ± 3.9	^a^ 35 ± 2.9	^ab^ 50.5 ± 1.5	^a^ 21.0 ± 1.6	^a^ 4.1 ± 0.2

## Data Availability

Data are provided in this article. Data will be provided on request.

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
