# Peer review of "Silicon-Selenium Interplay Imparts Cadmium Resistance in Wheat through an Up-Regulating Antioxidant System"

_ijms, 2023, doi:10.3390/ijms25010387_

Round 1
Reviewer 1 Report
Comments and Suggestions for Authors
The submitted manuscript explores the combined effect of Si and Se application on wheat's tolerance to Cd. Plant growth and chemical parameters (incl. elemental and enzyme concentrations) were measured. Below are some concerns that need to be further considered and better explained in the revised version.
-
Cd dose. The level of exposure to Cd is 10 uM, which translates into a concentration of over 1 mg/L. This Cd concentration can be rarely encountered in pore water in arable soils and is much higher than environmentally relevant levels. There should be a clear statement about this, as well as an explanation of how the results obtained at this high level of Cd exposure can be related to wheat production systems.
-
Se-Si interaction. As noted in the Introduction, the novelty of the study lies mainly in an examination of the combined effect of Se and Si on wheat. There should be a subsection in the Discussion dedicated to this point.
-
Mechanism of Cd tolerance (section 3.4). This study deals with chemical data, and there is no biological evidence (e.g., molecular data) to show how plants cope with Cd toxicity. As such, this part is speculative and not convincing.
-
Statistics issues. In Figure 1 for example, many factors (cell fraction, Cd exposure, Se and Si supply) were mixed to conduct a one-way (i.e. one factor) ANOVA and Duncan's post-hoc multiple comparisons. This is statistically incorrect.
In addition, there are some formulation issues:
1) It is important that each figure/table stands alone, so all abbreviations should have their full names in the caption.
2) In Table 1 and other tables, an asterisk marks statistical difference compared to which treatment? In addition, there are no asterisks in the Figures.
3) Figure 3 contains repetitive information (upper and lower half). What is "relative uptake"?
4) The localization maps in Figure 4 are not presented in the Results nor discussed.
Author Response
Thanks for appreciating and critically revieweing the manuscript. Togather with your suggestion, we hope that the revised manuscript is much improved and acceptable to the worthy reviewer. Thanks

Reviewer 2 Report
Comments and Suggestions for Authors
According to me, the manuscript entitled “Silicon-Selenium Interplay Imparts Cadmium Resistance in Wheat through Up-regulating Antioxidant System” promise to be interesting. The authors put a great deal of work into performing hydroponic experiments, estimation of antioxidative response of wheat plants, visualizations of ROS in root tissues, determination of subcellular distribution of Cd, as well as many other analyses. Manuscript contains information that is worthy of publication. A special achievement is the development of the mechanism of the protective effect of Se and Si on wheat growth. However, corrections should be made to make the presented manuscript more precise and understandable to readers Also, editing errors should be corrected and some passages should be made more detailed.
The manuscript is divided into clearly defined sections and contains all required chapters (Abstract, Keywords, Introduction, Results, Discussion, Materials and Methods, and Conclusions). All figures and tables are useful but not accurately cited in the text (e.g. line 406 – Table S1, while Tables S3, S4 and Figure S1 are not cited at all).
Title is informative and arouses the reader's interest. Abstract is concise. It defines the purpose of the research and the principal results. This abstract is able to function independently of the text of the manuscript. I suggest changing “removal in wheat” to “accumulation in wheat” (line 11). The sentence raises my doubts “Afterwards, the treatments including 0.2 mmol Si and 1.5 μmol Se were applied as a soil and foliar application, respectively.” (lines 16-17). The Authors do not mention anything about this in the methodology. Please also look at line 82: is the term "soil experiment" correct?
Introduction provide an adequate background for the objectives of the work and it is interesting literature survey. According to me, “including prostate, cancer, bone deformities” (line 50) should be changed to “including prostate cancer, bone deformities”. Also, “To avoid Cd food chain contamination, selenium (Se) and silicon (Si) have received much importance in recent years [6].” (lines 56-57) should be changed to “To avoid food chain contamination with Cd, using selenium (Se) and silicon (Si) have received much importance in recent years [6].”
Description of results is connected with illustrations (1 table and 5 figures), whereas supplementary materials are partially used. This chapter needed to be prepared more carefully. The authors should verify the description of statistically significant and non-significant changes, e.g. lines 94-95. The authors wrote about significant changes in biomass and shoot length, referring to Table 1. However, the data shown in this table do not show these significances (* indicates the significance of changes). Moreover, the description of changes in specific parameters does not provide information about which series they are compared to, e.g. sentences “Results of photosynthesis activity and transpiration indicated a significant reduction of 1.2– folds in Cd-stressed plants. Which were significantly increased by 1.3– and 1.4– folds, respectively.” (lines 101-103). This makes me lose trust in authors.
How was the tolerance index calculated? The authors do not mention this in the methodology.
How many replicates were there? 3 or 4?
Lines 121-124 “The data indicated dual strategies induced by Si and Se for Cd removal and stress management by wheat plants. Si-induced Cd removal (Fig. 1a,c,e). Whereas, Se induced Cd tolerance by regulating the antioxidant defense system (Fig. 4).” This fragment fits into the discussion.
Figure 1. Please correct the figure caption. In my opinion, figure 1b, 1d, 1f are unnecessary, because do not add anything to the description. The sentence "(*) represents statistical difference at p>0.05." is false.
Lines 130-133 This fragment fits better into chapter 2.2.1 than 2.2.2.
Figure 2. Please correct the figure caption. What is the difference between figures 2c and 2e and between 2d and 2f. What does TI mean?
Figure 4. Please correct the figure caption.
Figure 5. Please correct the figure caption.
Materials and Methods contains sufficient details to permit the readers understanding research methods. However, this chapter also contains parts that need to be corrected. In my opinion, the research project presented in this way lacks a series treated only with silicon. It is also possible to rearrange the data presentation plan and provide only series such as: 1) control, (2) Cd, (3) Cd + Se; (4) Cd + Si; (5) Cd + Si + Se.
Line 392 “0.2X nutrient solution” - what does “X” mean?
Lines 392-395 The composition of basal nutrients solution given in this fragment is different than in Table S1. Which one is real? uM or mM?
Line 396 “with 0.5X” - what does “X” mean?
Line 398 “7 treatments with 4 replicates” - this is the real replication count?
Line 402 “a pinch of NaCl” - this is not a scientific term.
Line 406 “(Table S1)” - bad table citation.
Line 414 “3 times rinsing in DI and water” - In my opinion it should be “3 times rinsing in DI water”
Line 465 “(10×)” What does it mean?
Line 491 “extracted by centrifugation” - centrifugation is not used for extraction.
Lines 512-513 “at 595 nm through an atomic absorption spectrophotometer” - this is not correct information.
According to me, this manuscript needs the major revisions.
Author Response

(The authors gave the same response as above.)

Reviewer 3 Report
Comments and Suggestions for Authors
The manuscript with the title Silicon-Selenium Interplay Imparts Cadmium Resistance in wheat through Up-regulating Antioxidant System was proposed for publication. The results are interesting, but, in order to improve the quality of the manuscript, I have some comments:
1. Page 1. Lines 32-34. Something is missing in the phrase. Please check it.
2. Page 1. Line 40. …is between 0.01 and 1 mg/kg…
3. Page 2. Lines 55-56. Some information about Cd in particular in wheat would be necessary. Also some information about wheat and wheat-products consumption around the world would be useful.
4. Page 14. Line 440. Described by Wu et al. [14].
5. Page 15. Line 477. Mention the equipment for the centrifugation and the producer.
6. Page 15. Line 478. Mention spectrophotometer type and the producer.
7. Page 15. Line 485. Mention the equipment for the homogenization and the producer or add the term “manually” if the homogenization was done in this manner.
8. Page 15. Line 486. Mention the equipment for the centrifugation and the producer.
9. Page 15. Line 491. Mention the equipment for the centrifugation and the producer.
10. Page 15. Line 495. Mention the water bath type and the producer.
11. Page 16. Line 502. Mention the equipment for the centrifugation and the producer.
12. Page 16. Line 506. Mention spectrophotometer type and the producer.
13. Page 16. Line 513. Mention spectrophotometer type and the producer.
Also, I have 2 major comments:
1. Figures should be self-readable, thus I recommend to introduce all the abbreviations used in the figure in a short legend or as supplementary text below the figure.
2. The entire part of Results and Discussions should be restructured as there is a “not so good” correlation between the text and the figures. It is too difficult to follow the results obtained, thus I recommend to organize better this part (eg. Line 107 – figure 2a, Line 123 – figure 1 a,c,e, Line 124 – Figure 4, Line 159 - Figure 2b….)
Author Response

(The authors gave the same response as above.)

Round 2
Reviewer 1 Report
Comments and Suggestions for Authors
Many thanks to the authors for their revisions. It appears that the paper has been greatly improved and is now suitable for publication.
Author Response
Thank you so much for appreciating and accepting our revised manuscript for publication in IJMS. We greatly honor all your comments and suggestions that were very helpful in improving the understanding of manuscript.

Reviewer 2 Report
Comments and Suggestions for Authors
Thank you very much for responding to my comments and clarifying the ambiguities that appeared in the manuscript.
However, there are still 2 mistakes.
1. In the methodology, the authors wrote "The experimental design included 7 treatments with 4 replicates". However, the captions under the figures (Fig. 1, 2, 4, 5) inform that "The bars are means ± standard error of three replicates." In my opinion, it should be written "The bars are means ± standard error of four replicates"
2. In the methodology, the authors give the concentration of micronutrients in the solution as micromoles (uM), while in the supplement their concentrations are given as millimoles (mM). This needs to be improved.
Author Response
Thank you very much for responding to my comments and clarifying the ambiguities that appeared in the manuscript.
Response: Authors are thankful to the worthy reviewer for the extensive review. We have carefully revised the manuscript and hope that the revised draft will meet the criteria for publication in IJMS.
However, there are still 2 mistakes.
In the methodology, the authors wrote "The experimental design included 7 treatments with 4 replicates". However, the captions under the figures (Fig. 1, 2, 4, 5) inform that "The bars are means ± standard error of three replicates." In my opinion, it should be written "The bars are means ± standard error of four replicates"
Response: The authors are thankful for highlighting this mistake. The manuscript has been carefully revised for this mistake.
In the methodology, the authors give the concentration of micronutrients in the solution as micromoles (uM), while in the supplement their concentrations are given as millimoles (mM). This needs to be improved.
Response: Thanks for the comment, the units have been revised in the supplementary document for clear understanding.

Reviewer 3 Report
Comments and Suggestions for Authors
The manuscript is in a much improved form, so I recommend its publication, with the following mention:
Fig. 3. To mention at the end that the figure can be perfectly understood if it is viewed in color or in the online version. Printing the figure in black and white can make it more difficult to understand the details.
Author Response
The manuscript is in a much improved form, so I recommend its publication, with the following mention:
Response: The authors are grateful for the appreciation as well as for critical comments to improve the understanding and the quality of the manuscript.
Fig. 3. To mention at the end that the figure can be perfectly understood if it is viewed in color or in the online version. Printing the figure in black and white can make it more difficult to understand the details.
Response: We are thankfull to the worthy reviewer for nice sugession. That will surely help reader to understand the figure and results easily. The suggested statement has been added at the end of Fig. 3 caption.
“Figure 1. Pearson's correlation coefficients (r) between different metals and nutrients for possible interactions in root (a) and shoot (b), respectively, for relative uptake by wheat plant. Significance is calculated at p<0.05. The figure can be perfectly understood if it is viewed in color or in the online version.”
